# Bariatric Surgery Outcomes in Patients with Chronic Kidney Disease

**DOI:** 10.3390/jcm12186095

**Published:** 2023-09-21

**Authors:** Adriana Pané, Maria Claro, Alicia Molina-Andujar, Romina Olbeyra, Bárbara Romano-Andrioni, Laura Boswell, Enrique Montagud-Marrahi, Amanda Jiménez, Ainitze Ibarzabal, Judith Viaplana, Pedro Ventura-Aguiar, Antonio J. Amor, Josep Vidal, Lilliam Flores, Ana de Hollanda

**Affiliations:** 1Endocrinology and Nutrition Department, Hospital Clínic de Barcelona, Villarroel 170, 08036 Barcelona, Spain; claro@clinic.cat (M.C.); bromano@clinic.cat (B.R.-A.); ajimene1@clinic.cat (A.J.); ajamor@clinic.cat (A.J.A.); jovidal@clinic.cat (J.V.);; 2Centro de Investigación Biomédica en Red de la Fisiopatología de la Obesidad y Nutrición (CIBEROBN), Instituto de Salud Carlos III (ISCIII), 28029 Madrid, Spain; 3Nephrology and Kidney Transplantation Department, Hospital Clinic de Barcelona, 08036 Barcelona, Spain; amolinaa@clinic.cat (A.M.-A.); montagud@clinic.cat (E.M.-M.);; 4Endocrinology and Nutrition Department, Althaia Universitary Health Network, 08243 Manresa, Spain; 5Institut d’Investigacions Biomèdiques August Pi Sunyer (IDIBAPS)—Fundació Clínic per a la Recerca Biomèdica (FCRB), 08036 Barcelona, Spain; rolbeyra@recerca.clinic.cat (R.O.); jviaplana@idibell.cat (J.V.); 6Laboratori Experimental de Nefrologia i Trasplantament (LENIT), Centre de recerca biomèdica Cellex (CRB CELLEX), Fundació Clinic, Institut d’Investigacions Biomèdiques August Pi Sunyer (IDIBAPS), 08036 Barcelona, Spain; 7Obesity Unit, Gastrointestinal Surgery Department, Hospital Clínic de Barcelona, 08036 Barcelona, Spain; 8Centro de Investigación Biomédica en Red de Diabetes y Enfermedades Metabólicas Asociadas (CIBERDEM), 28029 Madrid, Spain

**Keywords:** chronic kidney disease, bariatric surgery, sleeve gastrectomy, Roux-en-Y Gastric Bypass, weight loss, weight regain

## Abstract

Obesity increases the risk of developing chronic kidney disease (CKD), which has a major negative impact on global health. Bariatric surgery (BS) has demonstrated a substantial improvement of obesity-related comorbidities and thus, it has emerged as a potential therapeutic tool in order to prevent end-stage renal disease. A limited number of publications to date have examined the beneficial effects and risks of BS in patients with non-advanced stages of CKD. We aimed to investigate the safety of BS in patients with CKD stages 3–4 (directly related or not to obesity) and both the metabolic/renal outcomes post-BS. A total of 57 individuals were included (n = 19 for CKD-group; n = 38 for patients with obesity, but normal eGFR [control-group]). Weight loss and obesity comorbidities resolution after BS were similar in both groups. Renal function (eGFR [CKD-EPI]) improved significantly at the 1-year follow-up: Δ10.2 (5.2–14.9) (*p* < 0.001) for CKD-group and Δ4.0 (−3.9–9.0) mL/min/1.73 m^2^ (*p* = 0.043) for controls. Although this improvement tended to decrease in the 5-year follow-up, eGFR remained above its basal value for the CKD-group. Noteworthy, eGFR also improved in those patients who presented CKD not directly attributed to obesity. For patients with CKD, BS appears to be safe and effective regarding weight loss and obesity comorbidities resolution, irrespective of the main cause of CKD (related or not to obesity).

## 1. Introduction

Obesity has become a major global health challenge and part of its costs are due to obesity comorbidities, including chronic kidney disease (CKD) [1,2,3,4]. Patients with severe obesity present distinct glomerular morphologic changes (glomerulomegaly, expansion and proliferation of mesangial matrix, podocyte hypertrophy, and glomerular sclerosis). These lesions are independent from other obesity-associated diseases, such as diabetes or hypertension. Altogether, these lead to a clinical pattern of proteinuria and progressive renal failure [3]. Moreover, obesity has been associated with a more rapid progression of CKD to end-stage renal disease (ESRD) in patients with pre-existing CKD [5]. It should also be highlighted that CKD has a major negative effect on global health, both as a direct cause of morbidity and mortality and as an important risk factor for cardiovascular (CV) disease [6].

Bariatric surgery (BS) in individuals with severe obesity is associated with significant and sustained weight loss, substantial improvement of obesity-related comorbidities, and increased life expectancy [7]. Therefore, it emerges as a potential therapeutic tool in order to prevent CKD progression to ESRD [8,9]. An improvement in albuminuria and proteinuria after BS has been extensively reported, supporting its beneficial effect on renal function [10,11,12,13]. However, a limited number of publications thus far have examined the beneficial effects of weight loss surgery in patients with CKD. In addition, the reported study groups were clearly dissimilar and yet not comparable, including patients with different stages of CKD (non-advanced stages to ESRD), and even patients who have had kidney transplantation with subsequent BS [5,14,15,16,17,18,19,20,21,22,23]. However, no study has directly assessed if the renal benefits of BS are exclusive of those patients with CKD directly related to obesity, without considering other etiologies. Moreover, there are renal risks related to BS which include perioperative acute kidney injury (AKI), with reports ranging from 2.9 to 8.5% [24,25,26].

Considering this background, our study primarily aimed at exploring BS safety in patients with CKD stages 3–4, and evaluating both renal and metabolic outcomes in the short (1 year) and mid-term (5 years) after BS. As a secondary aim, we assessed BS renal outcomes in patients with CKD not directly related to obesity.

## 2. Materials and Methods

We retrospectively evaluated all patients followed at the Obesity Unit of our institution and eligible for BS (laparoscopic Roux-en-Y Gastric Bypass [RYGB] or sleeve gastrectomy [SG]) between January 2005 to January 2018. During this study period, a total of 2298 BS were performed (revisional surgeries were not considered). Eligibility criteria for BS were age between 18–70 years and a body mass index (BMI) above 40 Kg/m^2^ or above 35 Kg/m^2^ in the presence of obesity-related comorbidities [27]. The technical aspects and the selection criteria for RYGB or SG at our institution have previously been reported [28].

The inclusion criteria for the study group (CKD-group) were the presence of CKD, defined as an estimated glomerular filtration rate (eGFR [CKD-EPI]) < 60 mL/min/1.73 m^2^ for at least 3 months, irrespective of the cause, and severe obesity eligible for BS. Patients with CKD stage 5 or on dialysis were excluded from the analysis. Patients in the CKD-group were paired 1:2 with patients who also suffered from obesity and were eligible for BS but had normal renal function (control-group). An eGFR > 60 mL/min/1.73 m^2^ and the absence of albuminuria were required for patients in the control-group. The matching criteria included year and type of BS, age, BMI, sex, and major comorbidities (diabetes and hypertension diagnosis as well as years of evolution). Using this propensity index score, 19 patients with obesity and CKD (CKD-group) were matched with 38 subjects without CKD but also entitled to BS (control-group).

In order to assess BS renal benefits according to CKD etiology, two subgroups were designed: obesity-CKD (n = 10) and non-obesity-CKD (n = 9). In the first group, patients with CKD and no other cause identified beyond obesity and its comorbidities were included. The second group comprised those patients with CKD in which a specific etiology had been reported: nephrectomy (n = 3), amyloidosis (n = 1), lupus nephritis (n = 1), and polycystic kidney disease (n = 2). In two patients, no specific cause was registered in their medical records. However, as renal function impairment was previous to the overweight/obesity diagnosis, they were classified in the non-obesity-CKD subgroup.

All subjects who fulfilled inclusion criteria and had no exclusion criteria were followed for a minimum of 5 years. In order to establish mortality and longer-term (>5 years) renal outcomes after BS, all subjects’ vital status, and creatinine levels and/or need for renal replacement therapy were revised until their last contact with the medical care system during the study period (end 1 June 2023), being the median follow-up of 11.3 (9.0–14.9) years.

### 2.1. Clinical and Laboratory Measures

Demographic (sex, age), anthropometric (height, weight), and both clinical and analytical data were recorded. The use of specific pharmacological treatments was also documented. The existence of heart disease, obstructive sleep apnea-hypopnea syndrome (OSAHS), hypertension (defined as taking antihypertensive drugs or repeated clinical systolic blood pressure ≥ 140 mmHg and/or diastolic blood pressure ≥ 90 mmHg), diabetes (diagnosed according to ADA criteria [29]) and dyslipidemia (defined as taking lipid-lowering drugs or LDL-cholesterol > 160 mg/dL) was registered.

Standardized assays were used to measure blood glucose, HbA1c, lipid profile, and serum creatinine; eGFR was obtained with the Chronic Kidney Disease-Epidemiology Collaboration equation (CKD-EPI) [30].

### 2.2. Metabolic and Kidney-Related Outcomes following BS

In order to assess metabolic outcomes, the improvement and/or resolution of obesity-related comorbidities and body weight trajectory following BS were recorded. Weight loss was expressed as the percentage of total weight loss (TWL) and the percentage of excess weight loss EWL) [31]. Weight regain (WR) was calculated as the percentage of maximum weight lost [100*(post-nadir weight − nadir weight)]/(pre-surgery weight − nadir weight) [32]. Kidney outcomes evaluation included the assessment of renal function through eGFR (CKD-EPI).

### 2.3. Hospitalization, Surgical Complications and Mortality

Both early and late surgical complications (<30 or >30 days after BS), as well as the number of deaths by cause, were assessed. The existence of AKI in the post-operatory period along with the need for hemodialysis was specifically revised. Hospitalization parameters, such as length of stay and surgical time were also recorded.

### 2.4. Statistical Analyses

Data are presented as median and 25th and 75th percentiles, mean ± standard deviation (SD) or number (%) unless otherwise indicated. The normal distribution of continuous variables was evaluated with the Shapiro–Wilk test in addition to normal P-P plots. Inter-group differences in quantitative variables were assessed using Student’s *t*-test or Mann-Whitney U, as appropriate. The chi-square test was used to evaluate between-group differences in qualitative variables. McNemar’s test was used on paired nominal data. Paired t-tests or Wilcoxon tests were used to evaluate within-group differences in quantitative variables.

IBM SPSS Statistics 23.0 (SPSS Inc.; Chicago, IL, USA) and STATA/IC 15.0 (StataCorp.; College Station, TX, USA) for Windows were used to perform the statistical analysis. The significance level was defined as a *p*-value < 0.05, with 2-sided tests.

## 3. Results

### 3.1. Subjects’ Characteristics

Participants’ basal characteristics are shown in Table 1. On average, patients were about 57 years-old and 70% were women. The median BMI at baseline was 45.3 Kg/m^2^. A total of 19 (33%) patients had type 2 diabetes (T2D) and 49 (86%), had hypertension. As expected, CKD and control groups were well-balanced regarding age, sex, BMI, major comorbidities, and type of BS. Dyslipidemia was more common in the CKD-group than in the control-group (84.2% vs. 47.4%, *p* = 0.008). However, the percentage of patients under statin treatment was similar (68.7% vs. 61.1%, *p* = 0.642).

In the CKD-group, most patients were classified as stage 3 (6 [31.6%] for stage 3a and 7 [36.8%] for stage 3b), and only 6 (31.6%) were classified as stage 4. CKD etiology was established according to medical clinical records. Obesity was the most frequently reported cause (31.5%), followed by a tie between hypertension (15.8%), nephrectomy (15.8%), and others (10.5%), including lupus nephritis and amyloidosis. Among those patients classified in the obesity etiologic division, renal biopsy was performed just in one case and informed as focal segmental glomerulosclerosis. Polycystic kidney disease accounted for 10.8% of CKD causes and T2D accounted for 5.3%. CKD etiology could not certainly be established in 2 patients (10.5%).

### 3.2. Metabolic and Kidney Related Outcomes

Participant’s metabolic and kidney-related outcomes at 1 and 5 years after BS are presented in Table 2 and Table 3, respectively.

The weight loss achieved following BS, both at the 1-year and 5-year follow-up, was comparable between groups and successful according to the classical Reinhold’s criteria (EWL > 50% and/or BMI < 35 Kg/m^2^). Five years after BS, WR was similar, but with a wide range of variation: 16.2 (6.3–28.2)% vs. 19.7 (11.7–25.7)% for the CKD vs. control-group (*p* = 0.439). It has to be mentioned that three subjects in the control-group required revisional surgery (conversion of SG to RYGB) because of gastroesophageal reflux disease (GERD). None of the patients included in the CKD-group required revisional surgery.

Attending obesity comorbidities, in the CKD-group the prevalence of hypertension remained mainly unchanged following BS, but the number of anti-hypertensive drugs could be reduced. The percentage of T2D resolution at 1 year following BS [33] was similar in both groups (42.9% for CKD-group and 50% for controls; *p* = 0.764). The metabolic improvements reached at 1-year follow-up remained stable at the 5-year re-evaluation (Table 3). Regarding renal function, at the 1-year follow-up, eGFR clearly upgraded: Δ10.2 (5.2–14.9) (*p* < 0.001) for CKD-group and Δ4.0 (−3.9–9.0) mL/min/1.73 m^2^ (*p* = 0.043) for control-group. Although this improvement tended to decrease, eGFR remained above its basal value for the CKD-group: Δ3.8 (−1.7–9.7) mL/min/1.73 m^2^ (*p* = 0.031). Only one patient required hemodialysis before the completion of the 5-year follow-up period.

Figure 1 shows all the details regarding eGFR evolution post-BS in the longer-term. Close to the 5-year follow-up, two patients required renal replacement therapy (5.7- and 5.5-years post-BS, respectively). Specifically, one of them was placed on the national kidney transplant waiting list; the other one was excluded because of the coexistence of prostatic adenocarcinoma. Except for these two cases, eGFR remained stable for both study groups until the 10-year follow-up. After this period, eGFR tended to decrease without reaching a value inferior to 30 mL/min/1.73 m^2^ in any case.

As depicted in Figure 2, when assessing eGFR depending on the CKD stage, subjects classified in the higher CKD stages tended to worsen in the long follow-up. In fact, both patients requiring hemodialysis in the extended follow-up (>5 years) had been classified in CKD stage 4 baseline.

When specifically assessing BS benefits on renal function regarding the main etiology of CKD, eGFR improved at the 1-year follow-up in both subgroups. For the obesity-CKD group (n = 10), baseline eGFR increased from 48.6 (39.2–52.6) to 53.34 (46.7–57.9) mL/min/1.73 m^2^ at the 1-year follow-up (*p* = 0.009); similarly for the non-obesity CKD group (n = 8 since one patient died in the first follow-up), from a baseline value of 28.6 (25.4–32.5) to 40.6 (33.0–47.9) mL/min/1.73 m^2^ (*p* = 0.017).

At the 5-year follow-up, eGFR remained stable for the obesity-CKD division (n = 9 since one patient died in the second follow-up; 54.1 [48.0–54.4] mL/min/1.73 m^2^ vs. 1-year eGFR, *p* = 0.594). However, eGFR slightly deteriorated in the non-obesity CKD subgroup (n = 8; 35.2 [29.4–40.1] mL/min/1.73 m^2^ vs. 1-year eGFR, *p* = 0.036). As it has been previously stated, one male patient in the obesity-CKD division (obesity comorbidities) and one male patient in the non-obesity CKD subgroup (polycystic kidney disease) required hemodialysis in the extended follow-up at ages 60.5 and 67.7, respectively. Further details concerning the characteristics of the CKD subgroups can be found in Table 4.

### 3.3. Post-Operative Complications and Mortality

No major clinically significant differences regarding length of stay, surgical time, or postoperative complications were observed between the study groups. Two patients presented AKI, but none of them needed hemodialysis in the recent post-operative period (<30 days). All the details are presented in Table 5.

Regarding mortality, one patient in the control group (at age 73), and five patients in the CKD-group (at ages 27, 56, 61, 73, and 80) died. The most frequent causes were CV disease (myocardial infarction) and septic shock, being the source of infection the gastrointestinal tract and jugular endovascular catheter, respectively.

The median interval between the initial surgery and death was 2.9 (0.6–9.9) years in the CKD-group and, 12.1 years in the control-group. One patient in the CKD-group died shortly after having initiated renal replacement therapy (3 months from BS) because of an endovascular catheter infection.

## 4. Discussion

Our study primarily aimed to assess BS safety along with long-term renal and metabolic outcomes in patients with CKD. We have shown that BS appears to be safe in patients with CKD (stages 3–4). In fact, these patients are the ones whose renal function is susceptible to recovering after BS. Additionally, our results point out that BS is effective for achieving/maintaining weight loss and for improving obesity comorbidities independently of the presence of CKD. Although previous authors have also approached these objectives, this is the first long-term follow-up study applying a propensity-score matching method so as to correct for sample selection bias. In addition, it specifically considers BS renal and metabolic benefits attending CKD etiology in a binary approach (directly related or not to obesity).

Current evidence supports the use of BS to treat patients with obesity and CKD. Actually, several observational studies have analyzed kidney outcomes in patients with or without CKD who undergo BS [9,34,35,36]. Overall, results have consistently shown that BS is associated with slower eGFR decline [8,9,37,38,39]. In line with these findings, patients in our cohort experienced an improvement in eGFR in the short and medium term. Furthermore, it should be stressed that this improvement was irrespective of the main cause of CKD. Multiple mechanisms may be involved in this improvement. Recent studies point to a connection between lipotoxicity and CKD. Moreover, adipose tissue depots directly related to the kidney (renal sinus fat) might also play a role [39].

The last guidelines for the management of CKD emphasize the importance of controlling those risk factors associated with CKD progression such as hypertension, diabetes, and dyslipidemia, which are also widely recognized classical CV risk factors [40]. BS has demonstrated a marked reduction in CV morbidity/mortality and a significant increase in life expectancy [41,42,43] and thus, it should be considered as a useful, safe, and valid therapeutic tool in the CKD setting.

In agreement with previous observational studies in the field [14,36,37], we have shown an improvement of hypertension in the CKD-group reflected by a decreased need of anti-hypertensive medications at both the 1-year and 5-year follow-up. However, a resolution of hypertension following BS could not be demonstrated. Regarding T2D, although its prevalence remained almost the same at 1-year follow-up for patients in the CKD-group, HbA1c levels clearly improved. A similar situation occurred when considering the prevalence of dyslipidemia and the laboratory lipid profile. The levels of total cholesterol, non-HDL-cholesterol, and triglycerides improved, without any significant change in the use of statins. It must be underlined that these advances in both glycemic and lipid control persisted at the 5-year follow-up. Previous works and metanalyses have also reported BS effectiveness in order to achieve significant weight loss in patients with CKD, both in the short and medium-term. Our results reinforce these aforementioned findings and extend them to the long term [44,45].

A relatively small number of studies to date have tested the impact of renal function on the early postoperative outcomes following BS, suggesting an increased risk for complications in patients with more advanced CKD stages, especially with ESRD, due to the inherent associated negative health consequences of an additional concomitant chronic disease [5,23,46,47]. Although a higher rate of complications was also observed in our cohort, it did not result in an increased length of stay or what is more relevant, it did not have a negative impact on the immediate postsurgical mortality rate. Nevertheless, and as it could be expected, the presence of AKI after BS was only present in the CKD-group [24,25,26]. Renal replacement therapy was not needed immediately after BS, but 3 months after the surgical procedure one patient started on hemodialysis. Unfortunately, the patient suffered an endovascular catheter infection because of multidrug-resistant bacteria and died shortly after.

It should be strengthened that although the mortality rate was clearly higher in the CKD group, it was not an immediate consequence of BS. It is well-known that life expectancy is reduced for all levels of renal function below an eGFR of 60 mL/min/1.73 m^2^. Actually, CKD has been referred to as a silent and poorly known killer [48]. This higher risk of premature death is principally related to an increase in CV morbidity, as observed in our cohort [49].

Our study has strengths and limitations. The exhaustive short (1-year) and mid-term (5-year) follow-up, along with the comprehensive clinical evaluation (both renal and metabolic outcomes), of patients with CKD eligible for BS is one of its key strengths. Moreover, it includes a well-paired group of controls attending BMI, sex, age, main comorbidities, and year and type of BS. Finally, it precisely assesses BS renal benefits according to CKD etiology.

However, limitations should also be acknowledged. Firstly, the main drawback of our study is its small sample, especially attending subgroups (CKD directly related or not to obesity). Consequently, extrapolating our results to other populations has to be taken with great caution. Although the exact prevalence of obesity among patients with CKD can vary widely based on the criteria used to define both obesity and CKD, in Spain, it has been set at around 22.6% [50]. Therefore, a higher percentage of patients with CKD and obesity eligible for BS could be expected. Nevertheless, the specific characteristics of our Renal Unit, in which highly specialized dietitians offer specific dietary advice through regular visits might have contributed to a lower proportion of severe obesity and thus, explain our relatively small sample. Secondly, our control-group was not the most suitable one as it consisted of subjects with obesity entitled to BS, but normal kidney function. Although we aimed to include individuals with CKD and obesity who were also candidates for BS, but refused it, the number of eligible individuals was clearly insufficient, and the existence of multiple confounders (age, sex, comorbidities) could not be ruled out. Therefore, we opted to apply a propensity index score. Thirdly, regarding the assessment of renal function, the CKD-EPI equation with creatinine rather than cystatin C was used, thus possibly affecting the precision of eGFR. As cystatin C is not routinely collected in our institution, using the CKD-EPI was considered the most accurate choice [15]. Other important indicators of kidney function, such as proteinuria, were not widely available in our cohort. Nevertheless, the existence of proteinuria was carefully revised and excluded before including any patient in the control-group. Finally, the CKD etiology was established according to the available medical records. As a result, misclassifications cannot certainly be discarded.

## 5. Conclusions

Weight loss surgery in patients with CKD improves short-, medium-, and long-term kidney outcomes as well as obesity comorbidities. In addition, its benefits seem irrespective of CKD main cause considered in a binary approach (related or not to obesity). Consequently, BS could be considered as a renoprotective intervention in patients with pre-existing CKD and it should not be discouraged depending on CKD etiology. Nevertheless, these potential benefits must always be counterbalanced with eventual adverse events in multidisciplinary teams. Additional surgical risks should always be considered beforehand in order to be prepared and therefore avoid a negative impact not only on hospital length of stay but also complications and survival rate.

## Figures and Tables

**Figure 1 jcm-12-06095-f001:**
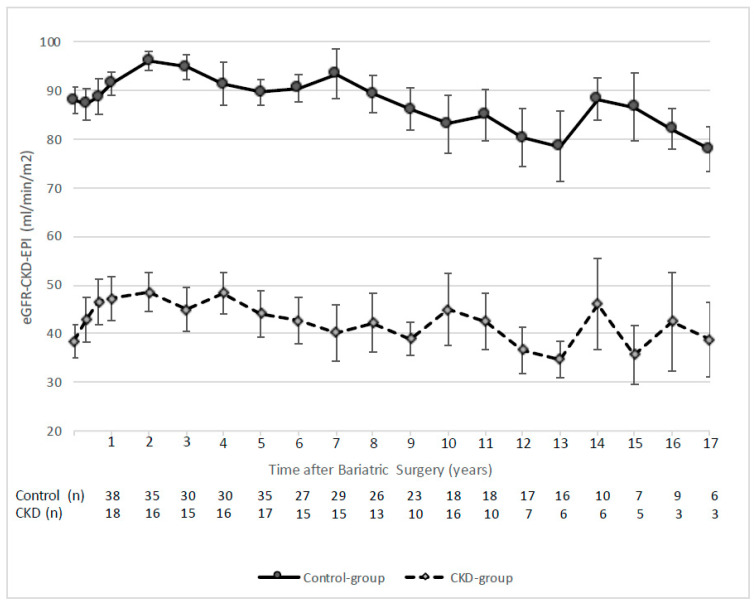
eGFR evolution after bariatric surgery for study groups. Data are shown as mean ± standard error.

**Figure 2 jcm-12-06095-f002:**
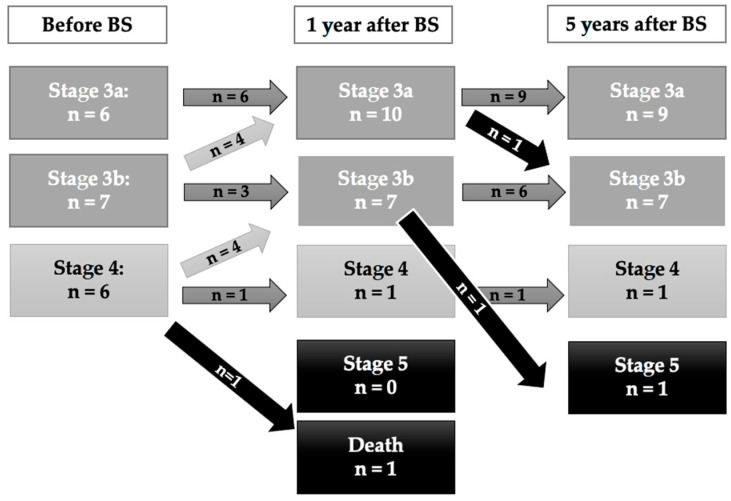
CKD stages for the CKD-group during the 5-year follow-up.

**Table 1 jcm-12-06095-t001:** Baseline characteristics of the CKD and control groups.

	CKD-Group (n = 19)	Control-Group (n = 38)	*p*-Value
**General clinical characteristics**			
Sex (female), n (%)	12 (63.2)	828 (73.7)	0.413
Age (years)	54.9 (53.1–59.9)	56.1 (53.9–58.8)	0.509
Weight (Kg)	118.2 ± 18.4	122.8 ±18.9	0.381
BMI (Kg/m^2^)	46.2 ± 8.2	47.8 ± 6.1	0.398
**Renal function**			
Serum creatinine (mg/dL)	1.50 (1.4–2.5)	0.77 (0.7–0.9)	<0.001
eGFR-CKD-EPI (mL/min/1.73 m^2^)	38.4 ± 15.0	88.0 ± 16.3	<0.001
**Comorbidities**			
Hypertension, n (%) Duration (years) n. Anti-hypertensive drugs	17 (89.5) 10 (7–14) 2 (2–3)	32 (84.2) 10 (6–15) 2 (1–2)	0.590 0.768 **0.003**
Diabetes, n (%) Duration (years) n. Oral hypoglycemic agents	7 (36.8) 6 (4–16) 1 (0–1)	12 (31.6) 8 (4–12) 1 (1–1.5)	0.691 0.865 0.448
Dyslipemia, n (%) Statins, n (%)	16 (84.2) 11 (68.7)	18 (47.4) 11 (61.1)	0.008 0.642
OSAHS, n (%)	4 (21.1)	16 (43.2)	0.101
Heart disease, n (%) Ischemic Hypertensive Valvular	3 (15.8) 2 (66.7) 1 (33.3) 0 (0)	5 (13.2) 2 (40.0) 2 (40.0) 1 (20.0)	0.787 0.465 0.850 0.408
**Laboratory results**			
Glucose (mg/dL),	114 (92–124)	101 (115–140)	0.397
Hemoglobin A1c (%)	6.4 (5.5–6.8)	5.9 (5.5–6.5)	0.477
Total cholesterol (mg/dL)	204.5 ± 41.5	204.4 ± 34.7	0.988
HDL-cholesterol (mg/dL)	44 (39–54)	47 (36–53)	0.813
LDL-cholesterol (mg/dL)	122.3 ± 35.0	130.7 ± 29.7	0.358
Triglycerides (mg/dL)	168 (115–217)	118 (90–171)	0.077
**Type of bariatric surgery**			
SG Surgical time (min) RYGB Surgical time (min)	13 (68.4) 90 (60–110) 6 (31.6) 102.5 (100–115)	25 (65.7) 75 (60–85) 13 (34.2) 120 (87–130)	0.843 0.413 0.843 0.597

Data are shown as n (%), median (Q1–Q3), or mean ± standard deviation. *p*-values for intergroup comparisons. BMI: body mass index; eGFR: estimated glomerular filtration rate; min: minutes; N: number; OSAHS: obstructive sleep apnea-hypopnea syndrome; RYGB: laparoscopic Roux-en-Y Gastric Bypass [RYGB]; SG: sleeve gastrectomy.

**Table 2 jcm-12-06095-t002:** 1-year follow-up post-surgical characteristics of study groups.

	CKD-Group (n = 19)		*p*-Value ^†^	Control-Group (n = 38)		*p*-Value ^†^	*p*-Value ^††^
	Baseline	1 Year after Surgery ^¶^		Baseline	1 Year after Surgery		
**General clinical characteristics**							
BMI (Kg/m^2^)	46.0 ± 8.4	30.5 ± 4.6	**<0.001**	47.8 ± 6.1	32.1 ± 4.6	<0.001	0.810
EWL (%)	---	76.6 ± 16.1	---	---	70.6 ± 15.8	---	0.206
TWL (%)	---	32.9 ± 6.2	---	---	32.8 ± 7.1	---	0.970
**Renal function**							
Serum creatinine (mg/dL)	1.5 (1.4–2.2)	1.3 (1.1–1.9)	**0.001**	0.8 (0.7–0.9)	0.7 (0.7–0.8)	**0.037**	**<0.001**
eGFR [CKD-EPI] (mL/min/1.73 m^2^)	39.8 ± 14.1	49.8 ± 16.6	**<0.001**	86.9 ± 16.3	91.4 ± 14.6	0.092	**0.003**
**Comorbidities**							
Hypertension, n (%) n. Anti-hypertensive drugs	16 (88.9) 2 (2–3)	14 (77.8) 2 (1–2)	0.500 **0.018**	32 (84.2) 2 (1–2)	17 (44.7) 1 (1–2)	**<0.001** **0.008**	**0.014** **---**
Diabetes, n (%) Insulin use, n (%) Oral hypoglycemic agents, n (%)	7 (38.9) 2 (11.1) 1 (0–1)	4 (22.2) 1 (5.5) 0 (0–0.5)	0.250 1 0.157	12 (31.6) 6 (15.8) 1 (1.0–1.5)	6 (15.8) 2 (5.3) 1 (1.0–1.0)	**0.031** 0.125 0.157	0.764 0.363 ---
Dyslipemia, n (%) Statins use, n (%)	16 (84.2) 11 (57.9)	11 (57.9) 8 (42.1)	0.125 0.453	18 (47.4) 11 (28.9)	10 (26.3) 5 (13.2)	0.008 **0.031**	0.681 0.670
OSAHS, n (%)	4 (22.2)	1 (5.6)	0.250	16 (43.2)	6 (16.2)	**0.002**	0.639
**Laboratory characteristics**							
Glucose (mg/dL)	114 (94–124)	91 (80–99)	0.007	115 (101–140)	88 (81–94)	**<0.001**	0.618
Hemoglobin A1c (%)	6.4 (5.6–6.8)	5.6 (5.2–5.9)	**0.001**	5.8 (5.5–6.3)	5.2 (5.0–5.6)	**<0.001**	0.994
Total cholesterol (mg/dL)	206.4 ± 41.8	185.4 ± 39.6	**0.017**	203.3 ± 34.8	194.9 ± 37.5	0.126	0.196
HDL-cholesterol (mg/dL)	44 (39–54)	51 (40–54)	0.222	47 (36–51)	53 (42–63)	**<0.001**	0.148
LDL-cholesterol (mg/dL)	124.6 ± 34.7	113.2 ± 39.4	0.119	130.8 ± 30.1	125.6 ± 31.4	0.252	0.443
Triglycerides (mg/dL)	172 (115–217)	101 (74–139)	**0.001**	118 (93–171)	83 (66–108)	**<0.001**	0.581
Non-HDL-cholesterol (mg/dL)	161.8 ± 42.2	136.1 ± 42.6	**0.005**	157.8 ± 30.2	143.0 ± 32.3	**0.002**	0.219

Data are shown as n (%), median (Q1–Q3), or mean ± standard deviation. *p*-values for intra **^†^** and inter-group **^††^** comparisons are reported. Inter-group change for serum creatinine and eGFR was adjusted for the corresponding baseline value. ^¶^ n = 18 since 1 patient died in the first year after BS. EWL: percent excess weight loss; TWL: percent of total weight loss; BMI: body mass index; eGFR: estimated glomerular filtration rate; N: number; OSAHS: obstructive sleep apnea-hypopnea syndrome.

**Table 3 jcm-12-06095-t003:** 5-year follow-up post-surgical characteristics of study groups.

	CKD-Group (n = 18)		*p*-Value ^†^	Control-Group (n = 38)		*p*-Value ^†^	*p*-Value ^††^
	1 Year after Surgery	5 Years after Surgery ^¶^		1 Year after Surgery	5 Years after Surgery ^¶^		
**General clinical characteristics**							
BMI (Kg/m^2^)	30.4 ± 4.8	32.0 ± 5.2	0.177	32.6 ± 4.2	34.4 ± 4.1	**0.001**	0.840
EWL (%)	---	68.1 ± 19.6	---	---	59.9 ± 14.3	---	0.093
TWL (%)	---	29.5 ± 10.6	---	---	28.2 ± 7.9	---	0.983
WR (%)	---	16.2 (6.3–28.2)	---	---	19.7 (11.7–25.7)	---	0.439
**Renal function**							
Serum creatinine (mg/dL) *	1.35 (1.13–1.9)	1.50 (1.08–1.8)	0.130	0.7 (0.7–0.8)	0.70 (0.6–0.8)	0.118	**0.004**
eGFR [CKD-EPI] (mL/min/1.73 m^2^) *	48.8 ± 16.5	44.4 ± 19.5	**0.028**	91.7 ± 14.9	90.0 ± 16.0	0.345	0.375
**Comorbidities**							
Hypertension, n (%) N. Anti-hypertensive drugs	13 (76.5) 2 (2–3)	14 (82.4) 2 (1–2)	1 0.317	17 (45.9) 2 (1–3)	19 (51.3) 1 (1–2)	0.727 0.936	0.800 ---
Diabetes, n (%) Insulin use, n (%) N. Oral hypoglycemic agents	4 (23.5) 1 (5.9) 0 (0–0.5)	3 (17.6) 1 (5.9) 1 (0–1)	1 1 ---	6 (16.2) 2 (5.4) 1 (1–1)	4 (10.8) 1 (5.3) 1 (0.5–1.5)	0.5001 ---	0.598 --- ---
Dyslipemia, n (%) Statins use, n (%)	11 (57.9) 8 (42.1)	10 (52.6) 6 (31.6)	1 0.625	10 (26.3) 5 (13.2)	13 (34.2) 10 (26.3)	0.375 0.125	0.314 0.506
OSAHS, n (%)	1 (5.9)	1 (5.9)	1	6 (16.2)	4 (10.8)	0.625	0.350
**Laboratory characteristics**							
Glucose (mg/dL)	114 (94–124)	96 (84–105)	0.887	116 (102–143)	90 (85–96)	0.148	0.753
Hemoglobin A1c (%)	6.4 (5.6–6.8)	5.9 (5.6–6.1)	0.499	5.8 (5.5–6.2)	5.5 (5.3–5.7)	**0.036**	0.980
Total cholesterol (mg/dL)	186.4 ± 40.6	195.9 ± 40.7	0.157	194.7 ± 37.7	195.6 ± 35.9	0.910	0.440
HDL-cholesterol (mg/dL)	51 (45–54)	55 (49–62)	**0.029**	52 (42–61)	57 (49–66)	**0.015**	0.973
LDL-cholesterol (mg/dL)	117.4 ± 39.3	118.3 ± 38.0	0.873	125.9 ± 30.9	118.6 ± 30.6	0.199	0.353
Triglycerides (mg/dL)	94 (74–132)	113 (66–135)	0.435	80.5 (71–101)	80.5 (70–112)	0.831	0.376
Non-HDL-cholesterol (mg/dL)	136.5 ± 43.9	139.4 ± 43.1	0.637	143.9 ± 32.7	136.7 ± 32.6	0.231	0.279

Data are shown as n (%), median (Q1–Q3), or mean ± standard deviation. *p*-values for intra **^†^** and inter-group **^††^** comparisons ^¶^ n = 17 for CKD-group as 2 patients died; n = 37 for control-group because of participants lost to follow-up. * n = 35 for control-group (missing data). EWL: percent of excess weight loss; TWL: percent of total weight loss; WR: percent of weight regain; BMI: body mass index; eGFR: estimated glomerular filtration rate; N: number; OSAHS: obstructive sleep apnea-hypopnea syndrome.

**Table 4 jcm-12-06095-t004:** Characteristics of the CKD subgroups.

	Obesity-CKD (n = 10)	Non-Obesity-CKD (n = 9)	*p*-Value
**General clinical characteristics**			
Sex (female), n (%)	6 (60.0)	6 (66.7)	0.764
Age (years)	56.2 (54.0–61.1)	54.6 (53.1–55.8)	0.191
Weight (Kg)	112.7 (103.0–127.0)	125.0 (108.0–127.7)	0.567
BMI (Kg/m^2^)	46.2 (39.2–53.3)	44.5 (39.4–47.5)	0.683
**Comorbidities**			
Hypertension, n (%)	8 (80.0)	2 (20.0)	0.156
Diabetes, n (%)	5 (50.0)	2 (22.2)	0.210
Dyslipemia, n (%)	8 (80.0)	8 (88.9)	0.596
OSAHS, n (%)	1 (10.0)	3 (33.3)	0.213
Heart disease, n (%)	3 (30.0)	0 (0.0)	0.073
**Baseline renal function**			
Serum creatinine (mg/dL)	1.4 (1.2–1.5)	2.2 (1.8–2.6)	**0.018**
eGFR [CKD-EPI] (mL/min/1.73 m^2^)	48.6 (39.2–52.6)	28.6 (25.4–32.5)	**0.011**
CKD stage, n (%) 3 4	9 (90.0) 1 (10.0)	4 (44.4) 5 (55.6)	**0.033**
**Mortality during follow-up**			
Number of deaths, n (%) CV disease Septic shock Renal disease progression Cancer	3 (30.0) 1 1 1 0	2 (22.2) 0 1 0 1	0.701

Data are shown as n (%), median (Q1–Q3). *p*-values for intergroup comparisons are reported. BMI: body mass index; CKD: chronic kidney disease; CV: cardiovascular; eGFR: estimated glomerular filtration rate; OSAHS: obstructive sleep apnea-hypopnea syndrome.

**Table 5 jcm-12-06095-t005:** Post-operative complications and mortality after bariatric surgery.

	CKD Group (n = 19)	Control-Group (n = 38)	*p*-Value
**Hospitalization parameters**			
Length of stay (days)	4.0 (2.9–6.2)	3.8 (2.9–4.2)	0.531
Surgical time (min)	95 (70–115)	80 (67–110)	0.525
**Post-operative complications**			
**Early (<30 days)** Major Anastomotic leak Respiratory failure Gastrointestinal hemorrhage Surgical site infection/hemorrhage Minor Peripheric catheter infection	4 (21.1) 1/4 (25.0) 1/4 (25.0) 1/4 (25.0) 1/4 (25.0) 0/4 (0.0)	4 (10.5) 2/4 (50.0) 0/4 (0.0) 1/4 (25.0) 0/4 (0.0) 1/4 (25.0)	0.281
**Late (>30 days)** Major Anastomotic leak Gastrointestinal hemorrhage Minor Anastomotic stricture Gastrointestinal ileus Acute non-biliary pancreatitis	3 (15.8) 1/3 (33.3) 1/3 (33.3) 0/3 (0.0) 0/3 (0.0) 1/3 (33.3)	2 (5.3) 0/2 (0.0) 0/2 (0.0) 1/2 (50.0) 1/2 (50.0) 0/2 (0.0)	0.185
**Acute kidney injury** Hemodialysis < 30 days	2 (10.5) 0/2 (0.0)	0 (0.0) ---	**0.042**
**Mortality during follow-up**			
Number of deaths Cardiovascular disease Septic shock Renal disease progression Cancer (renal)	5 (26.3) 1/5 (20.0) 2/5 (40.0) 1/5 (20.0) 1/5 (20.0)	1 (2.6) 1/1 (100) --- --- ---	**0.006**

Data are shown as n (%) or median (Q1–Q3). *p*-values for intergroup comparisons are reported.

## Data Availability

The data that support the findings of this study are available from the corresponding authors, A.D.H. and A.P., upon reasonable request.

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
