# Peer review of "Bariatric Surgery Outcomes in Patients with Chronic Kidney Disease"

_jcm, 2023, doi:10.3390/jcm12186095_

Round 1

Reviewer 1 Report

Dear Authors!

Congratulations for excellent and clinically very important work. 

Table 1 is maybee to large, especially for inconclusive date. Comparison both CKD group obese and non obese? both group are very overweight. I don t understand difference? both group after BS?

Author Response

We are so grateful for your comments.

We certainly agree with your point regarding “Table 1 is maybe too large for inconclusive data”, since redundant data was included, such as the percentage of patients with low HDL levels or the precise percentage of patients with LDL levels below 100 or 70 mg/dL. As these details do not add relevant information attending the paper's main aims, we have simplified Table 1 as suggested.

Thank you for making us aware that the presentation of the study groups may be quite confusing. Table 1 presents the baseline (before bariatric surgery [BS]) characteristics for both study groups. The CKD and the control groups share the same body mass index (BMI) as the definitory clinical difference between study groups is the presence or absence of an abnormal kidney function. BMI was one of the main criteria included in the propensity index score, therefore no statistically significant differences exist between the CKD and control groups.

We have rewritten the study-group inclusion and exclusion criteria in order to facilitate its interpretation (lines 78-84):

“The inclusion criteria for the study group (CKD-group) were the presence of CKD, defined as an estimated glomerular filtration rate (eGFR [CKD-EPI]) <60 ml/min/1.73 m2 for at least 3 months, irrespective of the cause, and severe obesity eligible for BS. Patients with CKD stage 5 or on dialysis were excluded from the analysis. Patients in the CKD-group were paired 1:2 with patients who also suffered from obesity and were eligible for BS, but had normal renal function (control-group). An eGFR >60 ml/min/1.73 m2 and the absence of albuminuria were required for patients in the control-group”.  

Reviewer 2 Report

Interesting paper regarding the complex relationships between obesity and CKD and the potential beneficial effects of bariatric surgery.

I have only 3 general remarks that in my opinion need to be mentioned/ addressed in text:

- after kidney transplatation graft survival is somehow related to BMI;

- it should be emphasized the possible role of BS in downstaging CKD

- the way Author deal with weight regain reflects the fuzzy picture in literature.  In bariatric practice Reinhold's criteria are usually the most employed, so I suggest to add also this option. If not considerable, mention why not.  

Author Response

Thank you for making us aware of these remarks in order to improve the quality and understanding of our manuscript.

  • We apologize for the poor background regarding kidney transplantation (KTx) graft survival and its association with BMI. However, last year our group already invested much effort in providing new data concerning bariatric surgery (BS) benefits for patients with obesity who were entitled to BS before or after KTx. Therefore, in the current manuscript we decided to just consider patients with CKD and to minimize references (and to avoid auto-citation) to the KTx context. Further details of our previous work can be consulted here: Pané A, Molina-Andujar A, Olbeyra R, Romano-Andrioni B, Boswell L, Montagud-Marrahi E, Jiménez A, Ibarzabal A, Viaplana J, Ventura-Aguiar P, Amor AJ, Vidal J, Flores L, de Hollanda A. Bariatric Surgery Outcomes in Patients with Kidney Transplantation. J Clin Med. 2022 Oct 13;11(20):6030. doi: 10.3390/jcm11206030. PMID: 36294351; PMCID: PMC9604744.

We completely appreciate your comment, since it empowers the importance of our previous work.

  • We certainly agree with your concerns regarding the necessity of emphasizing the possible role of BS in downstaging CKD. As suggested, we have reformulated the discussion and included a new significant and updated reference to this field.

Lines 259-267: “Current evidence supports the use of BS to treat patients with obesity and CKD. Actually, several observational studies have analyzed kidney outcomes in patients with or without CKD who undergo BS[9,34–36]. Overall, results have consistently shown that BS is associated with slower eGFR decline[8,9,37–39]. In line with these findings, patients in our cohort experienced an improvement in eGFR in the short and medium term. Furthermore, it should be stressed that this improvement was irrespective of CKD main cause. Multiple mechanisms may be involved in this improvement. Recent studies point to a connection between lipotoxicity and CKD. Moreover, adipose tissue depots directly related to the kidney (renal sinus fat) might also play a role[40]”.

Sandino J, Martín-Taboada M, Medina-Gómez G, Vila-Bedmar R, Morales E. Novel Insights in the Physiopathology and Management of Obesity-Related Kidney Disease. Nutrients. 2022;14(19):1-12. doi:10.3390/nu14193937

  • Following your advice, we have extended the details of body weight trajectories following BS. Therefore, not only the percentage of weight regain (WR) is mentioned in the main text, but also insufficient/sufficient weight loss according to the classical Reinhold’s criteria (1982). Further information can be also consulted in Table 2 and Table 3 (TWL: percent of total body weight loss and, EWL: percent of excess weight loss).

Lines 166-170: “The weight loss achieved following BS, both at the 1 and 5-year follow-up, was comparable between groups and successful according to the classical Reinhold’s criteria (EWL>50% and/or BMI<35 Kg/m2). Five years after BS, WR was similar, but with a wide range of variation: 16.2 (6.3-28.2)% vs. 19.7 (11.7-25.7)% for the CKD vs. control-group (p=0.439).”

Reviewer 3 Report

Thank the editors for the opportunity to review this paper. It is a very well written manuscript about the results of bariatric surgery in patients suffered from renal failure. I think this paper is very interesting, but some points are missing

Introduction: ok

Material and Methods: Which parametric and non-parametric statistical methods did the authors use? It should be added

Results: I believe that Table S1 and Figure S1 are relevant for this study and it should be in main manuscript, not the supplementary materials only. Especially I like Figure S1, which shows the changes

In my opinion the statements like 'Moving onto obesity comorbidities...' etc. should not be in manuscript. In the results we expect only facts

Table 4 is illegible. I think it would be better to give the percentage of the whole group in brackets, e.g. leak is 5% not 25%. Did the authors wonder what caused the high rate of complications? 

Discussion and conclusions: ok

Author Response

Thank you for the meaningful insights raised in this thorough revision.

  • We apologize for the lack of information regarding the precise statistical tests used for this study. We have amended the Material and Methods section as follows (lines 131-136): “Inter-group differences in quantitative variables were assessed using Student’s t-test or Mann-Whitney U, as appropriate. The chi-square test was used to evaluate between-group differences in qualitative variables. McNemar's test was used on paired nominal data. Paired t-test or Wilcoxon test were used to evaluate within-group differences in quantitative variables.”

  • As recommended, we have included Table S1 (now Table 2) and Figure S1 (now Figure 2) in the main document. Thank you for making us aware of the relevance of both Table S1 and Figure S1 in order to appropriately follow the study's main objectives. Furthermore, we have omitted the phrase “Moving onto” since is not the most appropriate one to use in the Results

  • We agree that Table 4 (now Table 5) is not easy to follow because multiple details are presented in a quite mixed way. In order to make it easier to track, we have modified the way in which the partial results are presented.

  • With regard to the higher rate of complications in the CKD-group, we have added our interpretation in the Discussion section as follows (lines 287-293): “A relatively small number of studies to date have tested the impact of renal function on the early postoperative outcomes following BS, suggesting an increased risk for complications in patients with more advanced CKD stages, especially with ESRD, due to the inherent associated negative health consequences of an additional concomitant chronic disease[5,23,47,48]. Although a higher rate of complications was also observed in our cohort, it did not result in an increased length of stay or what is more relevant, it did not have a negative impact on the immediate postsurgical mortality rate.”